# Structural Characteristic of the Arcuate Fasciculus in Patients with Fluent Aphasia Following Intracranial Hemorrhage: A Diffusion Tensor Tractography Study

**DOI:** 10.3390/brainsci10050280

**Published:** 2020-05-06

**Authors:** Hyeong Ryu, Chan-Hyuk Park

**Affiliations:** Department of Physical & Rehabilitation Medicine, Inha University School of Medicine, Inha University Hospital, Incheon 22332, Korea; allen121212@naver.com

**Keywords:** arcuate fasciculus, aphasia, intracranial hemorrhage, diffusion tensor image

## Abstract

This study investigated the relationship between the structural characteristics of the left arcuate fasciculus (AF) reconstructed using diffusion tensor image (DTI) and the type of fluent aphasia according to hemorrhage lesions in patients with fluent aphasia following intracranial hemorrhage (ICH). Five patients with fluent aphasia following ICH (three males, two females; mean age 55.0 years; range 47 to 60 years) and with sixteen age-matched heathy control subjects were involved in this study. The ICHs of patients 1 and 2 were located in the left parietal lobe and the left basal ganglia. ICHs were located in the left anterior temporal of patient 3, the left temporal lobe of patient 4, and the left frontal lobe of patient 5. We assessed patients’ language function using K-WAB (the Korean version of the Western Aphasia Battery) and reconstructed the AF using DTI. We measured DTI parameters including the fractional anisotropy (FA), tract volume (TV), fiber number (FN), and mean diffusivity (MD). All patients showed neural tract injury (the decrement of FA, TV, and FN and increment of MD). The left AFs in patients 1 and 2 were shifted from Broca’s and Wernicke’s territories. The destruction of Wernicke’s territory resulted in conduction or transcortical sensory aphasia in patients 3 and 4. The structural difference of the AF in patients following ICH in the left hemisphere was associated with various types of fluent aphasia.

## 1. Introduction

Human language function plays a role in the left hemisphere [1]. Aphasia has indicated an language impairment effect in the production and/or comprehension of speech and is one of the common sequelae following stroke in the left hemisphere [2,3]. Aphasia was shown in approximately 21–38% of acute stroke patients, and the major damaged lesion is the left perisylvian [4]. The arcuate fasciculus (AF) plays an important role for language in brain and, related with aphasia, is a neural fiber connected between Broca’s area for speech production and Wernicke’s area for comprehension and originates in the temporal lobe, passes around the Sylvian fissure, and projects to the frontal lobe [1,3,5]. Several studies have suggested that injury of the AF in the dominant hemisphere induced aphasia, and injury of the left AF has been reported to contribute to various language deficits, including non-fluent aphasia and fluent aphasia [3,6,7,8,9,10,11,12]. The estimation of the AF is important for the rehabilitative strategy and predicting the prognosis of language deficits [6]. In particular, lesion for fluent aphasia is located posteriorly in the cerebral hemisphere and fluent aphasia is composed of Wernicke’s aphasia, transcortical sensory, conduction and anomic aphasia [9,10,12,13,14]. 

Diffusion tensor tractography (DTT), derived from diffusion tensor imaging (DTI), has reconstructed the neural tracts and can visualize injury of the neural tracts which is not observed on conventional MRI (magnetic resonance image) [6,15,16,17,18]. Many studies have investigated injuries to the AF in the dominant hemisphere using DTI [3,5,6,8,19]. Although the previous study suggested the severity of aphasia in post-stroke patients according to lesion location and aphasia type using the Korean-Western Aphasia Battery (K-WAB), there have been no studies to assess the structural difference of the AF in the dominant hemisphere according to hemorrhage lesions in subacute stroke patients with fluent aphasia [4]. In this study, we attempted to investigate the relationship between the structural difference of bilateral AFs reconstructed using DTI and types of fluent aphasia in subacute stroke patients following intracranial hemorrhage (ICH) in the left hemisphere.

## 2. Materials and Methods

### 2.1. Subjects

Five patients suffered from speech and language impairment following ICH (three males, two female; mean age 55.0 years; range 47 to 60 years) and sixteen age-matched control subjects (four males; mean age 55.8 years; range 45 to 60 years) without past medical history were involved in this study (Table 1). The mean education period of patients was 11.8 ± 3.19 years. The inclusion criteria were as the follows: (1) first-ever stroke, (2) following ICH in the left hemisphere, (3) fluent aphasia on the Korean-Western Aphasia Battery (K-WAB), (4) no history of traumatic brain injury or psychiatric disorder, and (5) no oromotor dysfunction or language disorder before the current ICH. We measured the volume of hematoma using a picture archiving communication system (PACS) on T2-weighted MRI images with reference to the previous study [20]. This study protocol was performed retrospectively. According to the Declaration of Helsinki, the study was approved by Inha University Hospital Institutional Review Board (INHAUH-2019-10-014) on 17 October 2019.

### 2.2. Language Evaluation

The assessment of language deficit used the K-WAB (the Korean version of the Western Aphasia Battery) as the previous study [3]. The K-WAB is composed of oral language areas including spontaneous speech which consists of fluency and information contents, auditory comprehension, repetition, and naming [21]. The score for representative items is 20 of spontaneous speech, 200 of auditory comprehension, 100 of repetition, and 100 of naming. The reliability and validity of K-WAB are well established. Aphasia types were categorized using the previous study [14]. Additionally, we obtained the aphasia quotient (AQ), which represents fluency score, comprehension, repetition, and naming, and language quotient (LQ), which reflects reading, writing performance, auditory comprehension, and oral expression, from K-WAB [21,22]. AQ and LQ for the severity of aphasia were calculated using the formula suggested in the previous study [22]. These values ranged from 0 to 100th percentile and were divided into five severity criteria as follows; mild: 80–99th, mild to moderate: 60–79th, moderate: 40–59th, moderate to severe: 20–39th, severe: 0–19th [6]. 

### 2.3. Diffusion Tensor Image

DTI images were performed at approximately three or four weeks after onset using a 3.0 T GE Signa Architect MRI System (General Electric, Milwaukee, WI, USA). Image parameters were as the following conditions: 30 directions, and 72 contiguous slices, slice thickness = 2 mm, field of view = 240 × 240 mm, acquisition matrix of 128 × 128, b = 1000 mm^2^/s, Repetition time (TR) = 15,000 ms, and Echo time (TE) = 80.4 ms. The analysis of DTI used DTI studio software (www.mristudio.org, Johns Hopkins Medical Institute, Baltimore, MD, USA). For a reconstruction of the AF, we identified fibers passing through two regions of interest (ROIs). A seed ROI was placed in the lateral to the corona radiate and medial to the cortex on coronal slices, and a target ROI was selected in superior temporal gyrus on axial slices (Figure 1A). Fiber tracking conditions were a fractional anisotropy (FA) of < 0.2 and a turning angle of > 60°. The structure of the left AF was compared with the right AF. Based on Broca’s territory or Wernicke’s territory of the right AF on the coronal slice, we investigated the up or down movement of the left AF (Figure 1(Ba)). On the axial view, the shift of the left AF was observed compared with Broca’s area of the right AF (Figure 1(Ab)). In the control subjects, no difference between bilateral AFs was shown. DTI parameters including fractional anisotropy (FA), tract volume (TV), fiber numbers (FN), and mean diffusivity (MD) were measured. DTI parameters for fifteen control subjects were summarized in Table 2. Statistical analysis was conducted using SPSS software (version 26.0; SPSS, Chicago, IL, USA). Quantitative data for control subjects showed the means (standard deviations). For the differences between DTI parameters of each patient and those of the control group, we conducted an analysis using Bayesian independent t-test statistics [23]. A *p* < 0.05 was considered significant.

## 3. Results


*Patient 1*


A 60-year-old female with a history of hyperlipidemia was diagnosed with an ICH in the left parietal lobe (Table 1). She underwent conservative management. She suffered from language and speech impairment immediately after onset. The K-WAB indicated 27.5 scores (20.1%ile) of AQ and 22.3 scores (20.8%ile) of LQ at twenty-one days after onset. The severity of the aphasia was moderate to severe. The type of aphasia was Wernicke aphasia. The volume of hematoma was 27.44 mL. The left AF on DTI at twenty-one days after onset was thinner compared with control subjects and shifted down compared with the right AF. In the axial view, Broca’s territory of the left AF was moved posteriorly based on that of the right AF. The left AF showed a significant decrement of FA, TV, and FN values compared to those of control subjects. Additionally, the MD value increased significantly compared with those of control subjects (Table 2 and Figure 2A). However, DTI parameters of her right AF were not statistically significant compared with control subjects.


*Patient 2*


A 47-year-old male with an ICH in the left basal ganglia showed the right upper and lower extremities’ weakness and aphasia after onset (Table 1). He did not have a past medical history. Twenty-eight days after onset, although improvement in the right upper and lower limb weakness was shown, he suffered from aphasia. The characteristics of aphasia were 31.8 scores (23.2%ile) of AQ and 23.4 scores (21.7%ile) of LQ in K-WAB, and the aphasia type was Wernicke aphasia. The severity of the aphasia was moderate to severe. The volume of hematoma was 13.23 mL. DTI showed the thinner left AF and wider between Broca’s and Wernicke territory than that of a control subject. Wernicke territory was anteriorly longer than that of a control subject. Broca’s territory of the left AF was shifted posteriorly compared with that of the right AF in the axial view. The FA, TV, and FN values of the left AF were significantly decreased compared with those of the control subjects. However, there was no difference in MD value compared with the control subjects (Table 2 and Figure 2B). The DTI parameters of his right AF showed no significant difference compared with control subjects.


*Patient 3*


A 50-year-old male without past medical history underwent craniectomy for an ICH in the left anterior temporal lobe (Table 1). The patient complained of speech and language impairment immediately after onset. The K-WAB result showed 62.1 scores (51.0%ile) of AQ and 57.6 scores (58.3%ile) of LQ at about twenty-five days after onset. The severity of the aphasia was moderate, and the type of aphasia was transcortical sensory aphasia. Twenty-five days after onsest, DTI showed the discontinuation at Wernicke’s territory in the left AF with a posterior shift of Broca’s territory compared with the right AF. His volume of hematoma was 11.87 mL. The FA value of the left AF revealed significant decrement, while TV and MD values of the left AF did not show a significant difference compared with control subjects, while no difference between the right AF of the patient and control subjects was observed (Table 2 and Figure 2C).


*Patient 4*


A 60-year-old female with a history of hypertension underwent conservative management for an ICH in the left temporal lobe and complained of language and speech impairment immediately after onset (Table 1). The K-WAB indicated scores of 69.5 (73.6%ile) for AQ and 67.9 (81.8%ile) for LQ at twenty-one days after onset. The type of aphasia with a moderate severity of aphasia was conduction aphasia. Her volume of hematoma was 7.99 mL. The left AF in the DTI at twenty-one days after onset was thinner compared with control subjects and a few tracts at Wernicke’s area in the left AF with a posterior movement of Broca’s territory were shown. The left AF showed a significant decrement in FA value compared with control subjects. Additionally, MD, TV, and FN values revealed significant increment compared with those of control subjects (Table 2 and Figure 2D). There was no difference between the right AF of the patient and that of control subjects.


*Patient 5*


A 58-year-old male without past medical history was diagnosed as having an ICH in the left frontal lobe and corpus callosum genu (Table 1). He underwent conservative management at the acute phase. From the day of the onset, the patient suffered from speech and language impairment. The K-WAB indicated a 70.6 (75.1%ile) AQ score and 65.7 (79.6%ile) LQ score at twenty-eight days after onset. The type of aphasia with a moderate severity of aphasia was anomic aphasia. At twenty-eight days after onset, the volume of his hematoma was 11.73 mL and the left AF on DTI was thinner compared with control subjects and the tract at Broca’s area was posteriorly shifted. The left AF showed a significant decrement in TV and FN values compared with control subjects. On the other hand, the FA and MD values indicated no difference compared with those of control subjects (Table 2 and Figure 2E). Compared with the right FA of control subjects, his right AF showed no difference.

## 4. Discussion

This study investigated injury of the left AF in terms of DTI integrity and parameters according to ICH lesions in patients with fluent aphasia following ICH. Patients showed a different type of fluency aphasia (patient 1, Wernicke’s aphasia; patient 2, Wernicke’s aphasia; patient 3, transcortical sensory aphasia; patient 4, conduction aphasia; patient 5, anomic aphasia). With respect to DTI configuration, all left AFs showed structural change (discontinuation, narrowing or longer fiber, movement of the AF) around Wernicke’s territory and Broca’s territory compared to the control subjects. Two AFs (patients 1 and 2) indicated a structural change of the left AF such as shift up or down compared with control subjects or the right AF. Two AFs (patients 3 and 4) revealed the discontinuation of Wernicke’s territory, and three AFs (patients 2, 4, 5) exhibited a narrowing structure compared to the control subjects. DTI parameters including FA, TV, FN, and MD values of all patients indicated significant differences compared to the normal controls. 

FA values indicated degrees of directionality at a microscopic level and microstructural integrity of axons, myelin, and microtubules [16,24,25]. MD values represent the quantitation of water diffusion and an increase in MD values provides an indication of pathological change [15,16,18]. TV values reflect the number of voxels within neural tracts [16,18]. FN values represent the total fiber number in the neural tract [26,27]. Therefore, changes to these parameters indicate the presence of a neural injury [16,18,23,26]. As a result, our patients indicated injury of the left AF regardless of the hemorrhage lesion and these findings suggest that this injury reflects language impairment (Table 2).

Patients 1 and 2 with Wernicke’s aphasia showed a structural change to the left AF with a neural tract injury. Patient 1 indicated that Wernicke’s and Broca’s territory of the left AFs was shifted down. The damage localized to the inferior parietal lobe provides phonemic paraphasia without impairing, but our finding is different from previous findings (aphasia type of a patient: Wernicke aphasia) [28]. We demonstrated that the structural feature of the left AF plays a more important role in language impairment than the localized lesion. The parietal lobe ICH was responsible for movement and narrowing of the left AFs with neural tract injury. Additionally, in the previous study, the left basal glia lesion including caudate, putamen, internal capsule, and Globus pallidum caused a language deficit including Broca’s aphasia as well as Wernicke’s aphasia. In particular, putamen lesions with posterior extension represented a severe comprehension deficit and left internal capsule lesions showed comprehension and naming deficits [29]. Our result for patient 2 with hemorrhagic lesions in the putamen, Globus pallidum, and internal capsule is consistent with the previous result. The structure of the left AF using DTI in patient 2 was shifted up in Broca’s territory and down in Wernicke’s territory and long Wernicke’s territory. Thus, we suggested that the left basal ganglia ICH affected the structural change of AF, and this change contributed to Wernicke’s aphasia. Comprehensively, Wernicke’s aphasia is responsible for the change in microstructure of the left AF away from Wernicke’s and Broca’s area with neural tract injury. 

The previous study demonstrated that the AF was connected from the inferior frontal gyrus to the posterior superior temporal gyrus (STG), and the dorsal pathway and ventral pathway played a role in verbal repetition and auditory comprehension [2]. Referring to this evidence, we suggested that the shift of the left AF by the parietal lobe and basal glia ICH (patients 1 and 2) induced Wernicke’s aphasia, and the difference of comprehension score between them was related to the morphology of Wernicke’s territory. Thus, this shift of the tract through inferior frontal gyrus or posterior STG was associated with injury to the dorsal and ventral pathways, and these contributed to deficits of auditory comprehension and verbal repetition.

Damage to the anterior temporal lobe produced pure word deafness and transcortical sensory aphasia due to semantic system as well as temporal lobe lesion following stroke-induced conduction aphasia [28]. Patient 3 underwent ICH in the anterior temporal lobe, and the K-WAB result was transcortical sensory aphasia consistent with the previous study [28]. DTI reconstruction in patient 3 represented the discontinuation of Wernicke’s territory. The DTI finding in patient 4, also observed as conduction aphasia in the K-WAB, showed a few fibers in Wernicke’s territory. The shift in the left AF was not observed compared with Broca’s area of the right AF in patients 3 and 4. As a result, it could be suggested that transcortical sensory aphasia or conduction aphasia by damage to the temporal lobe resulted from injury to Wernicke’s territory without up or down movement of Broca’s area compared with the right AF. In other words, our findings demonstrate that temporal lobe ICH was directly associated with the disruption of Wernicke’s territory.

Finally, patient 5 with ICH in the left frontal lobe indicated anomic aphasia, because he preserved fluency, repetition, and comprehension relatively well, and showed a naming deficit. For language functions such a naming, a distributed network of neural regions in the left frontal, temporal, and parietal cortex was required [30]. Other studies demonstrated that one of the distributions of neural networks regarding word retrieval was the anterior cingulate cortex or frontal lobe, which is correlated picture naming [31,32]. Therefore, we suggested that anomic aphasia in our patients resulted from a posterior shift of Broca’s territory in the left AF by frontal lobe ICH due to the destruction of the network. However, because the damage of the frontal lobe is ascribed to Broca’s aphasia, further evaluation is required [28]. 

It was suggested that the prognosis of aphasia in post-stroke patients resulted from the structural change in the previous study [33]. This study showed that a right lateralization of AF led to poorer naming recovery. In constituent with the previous study, our patients (patients 1 and 2) found a lateralization and they showed Wernicke’s aphasia with poor naming function. However, while post-stroke patients in the previous study were in the chronic stage, our patents were in the subacute stage. Thus, further evaluation for long-term follow up images and the degree of lateralization is required. Due to the damage to the left AF in the dorsal pathway with Wernicke’s territory without the shift in that of the left AF compared with Broca’s area of the right AF, injury to the fiber at Wernicke’s territory by temporal lobe ICH represented transcortical sensory aphasia (patient 3) or conduction aphasia (patient 4) with impairment of auditory comprehension. ICH in the frontal lobe, including the inferior frontal gyrus, is responsible for anomic aphasia due to damage to the dorsal pathway (patient 5). In addition, all patients revealed neural tract injury of the left AF. The AFs reaching Broca’s area in all patients were shifted posteriorly regardless of lesions. We hypothesized that the posterior movement of the left AF resulted in the decrement in fluency score in K-WAB. Additionally, the previous study revealed asymmetries of bilateral AFs using the fiber density of bilateral AFs, volume of both tracts, and the number of voxels activated in the right and left hemisphere using functional brain MRI [34,35,36]. The asymmetry of the AF tract plays an important role in understanding language function [34]. However, this study did not show an asymmetry between bilateral AFs because this study performed the structural feature according to hemorrhagic lesion in post-stroke patients. Therefore, to identify the above hypothesis and relationship between the structural asymmetry and language function, further study is required.

To the best of our knowledge, this study is the first to investigate the structural change in the left AF in patients with fluent aphasia following ICH. The result shows that the destruction of Wernicke’s territory without the change in Broca’s area caused conduction or transcortical sensory aphasia due to ICH in the temporal lobe, and the parietal lobe or basal ganglia ICH contributed to Wernicke’s aphasia due to the shift and neural tract injury. However, some limitations of this study are as follows: (1) larger scale long-term studies are necessary to confirm our hypothesis; (2) because DTI interpretation depends on an operator, this leads to potential performance bias [16]; (3) because some studies demonstrated that language function was associated with location in the brain, the further study for the relation of the defined location using functional MRI and DTI is required [13,28,32]. (4) Due to the AF with large known inter-hemispheric differences, our research to compare between the left and right AFs using five patients is limited precisely to describe the anatomical difference between the left and right hemisphere [37]. 

## 5. Conclusions

In conclusion, as the previous study represented that the anatomical structure of the AF in language function was important, our findings are consistent [38]. Injury to the left AF is associated with aphasia, and the structural change in the left AF in patients with various types of fluent aphasia following ICH is found using DTI. In other words, the shift in the left AF away from Broca’s or Wernicke’s area with a neural tract injury induced severe aphasia such as Wernicke’s aphasia, and the disruption of the left AF in Wernicke’s territory with a neural tract injury led to conduction or transcortical sensory aphasia. Therefore, types of fluent aphasia depend on the structure of the left AF. DTI is useful to investigate the structure of the AF and predict types of aphasia.

## Figures and Tables

**Figure 1 brainsci-10-00280-f001:**
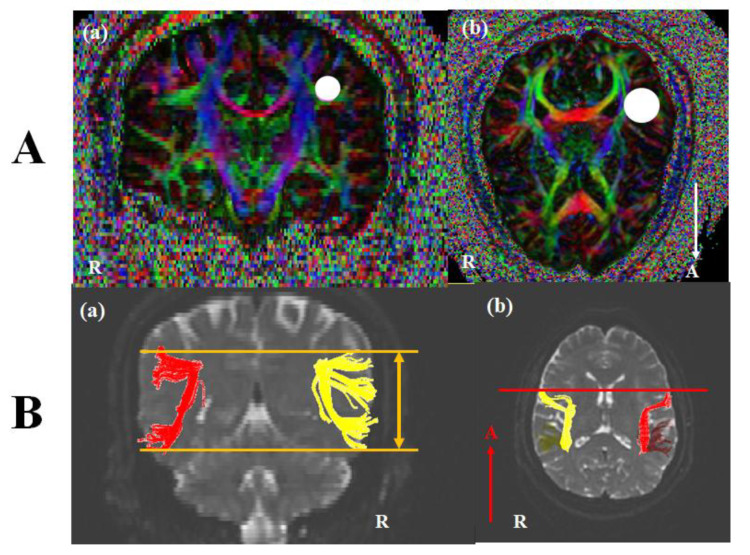
(**A**) Seed and target region of interest (ROI) of the AF. For the AF, (**a**) A seed ROI was placed in the lateral to the corona radiate and medial to the cortex on coronal slices (white circle), and (**b**) a target ROI was selected in superior temporal gyrus on axial slices (white circle). (**B**) Diffusion tensor image (DTI) of the AF. (**a**) Based on Broca’s territory or Wernicke’s territory of the right AF on the coronal slice, we investigated the up or down movement of the left AF (orange line). (**b**) Inn the axial view, the shift of the left AF was observed compared with Broca’s area of the right AF (red line). Note: AF, arcuate fasciculus; R, right; A, anterior.

**Figure 2 brainsci-10-00280-f002:**
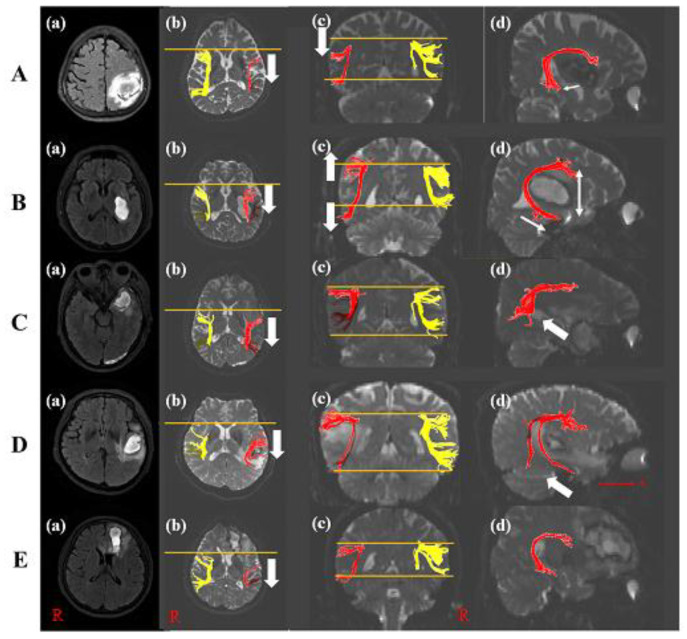
Diffusion tensor image of five patients. (**A**) T2-weighted MRI of patient 1 with the left parietal lobe ICH (**a**). The posterior shift (**b**) and down shift (**c**) of the left AF compared with the right AF was shown (white arrow). The disruption of Wernicke’s territory of the left AF appeared compared with that of control subject ((**d**), white arrow). (**B**) T2-weighted MRI of patient 2 with the left basal ganglia ICH (**a**). The posterior shift (**b**) and wide width (**c**) of the left AF compared with the right AF was shown (white arrow). The longer length of Wernicke’s territory of the left AF appeared than that of control subject ((**d**), white arrow). (**C**) T2-weighted MRI of patient 3 with the left anterior temporal lobe ICH (**a**). The posterior shift (**b**) and the disruption of Wernicke’s territory of the left AF appeared compared with that of control subject ((**d**), white arrow) without the difference in the structure compared with the right AF (**c**). (**D**) T2-weighted MRI of patient 4 with the left temporal lobe ICH (**a**). The posterior shift (b) and a few tracts of Wernicke’s territory of the left AF appeared compared with that of control subject ((**d**), white arrow) without the difference compared with the right AF (**c**). (**E**) T2-weighted MRI of patient 5 with the left frontal lobe ICH (**a**). Broca’s territory was shifted posteriorly (**b**) and showed thinner tracts compared with controls without the change of Wernicke’s territory ((**c**,**d**)). Note: MRI, magnetic resonance image; AF, arcuate fasciculus; ICH, intracranial hemorrhage; R, right; A, anterior.

**Table 1 brainsci-10-00280-t001:** Demographic data of patients and results of Korean-Western Aphasia Battery (K-WAB).

				K-WAB						
Patient	Sex/Age (Years)	Duration to DTI (Days)	Lesion Site	AQ (%ile)	LQ (%ile)	Spontaneous/20 (Fluency/10)	Comprehension/200	Repetition/100	Naming/100	Type
1	F/60	21 days	Left parietal lobe	27.5 (20.1)	22.3 (20.8)	9 (6)	75	4	6	Wernicke
2	M/47	28 days	Left basal ganglia	31.8 (23.2)	23.4 (21.7)	11 (7)	34	6	26	Wernicke
3	M/50	25 days	Left temporal lobe	62.1 (51.0)	57.6 (58.3)	13 (6)	129	73	43	Transcortical sensory
4	F/60	21 days	Left temporal lobe	69.5 (73.6)	67.9 (81.8)	15 (8)	140	59	70	Conduction
5	M/58	28 days	Left frontal lobe	70.6 (75.1)	65.7 (79.6)	12 (6)	143.5	90	71	Anomic

Note: K-WAB, the Korean-Western Aphasia Battery; F, female; M, male; AQ, aphasia quotient; LQ, language quotient.

**Table 2 brainsci-10-00280-t002:** Diffusion tensor image (DTI) parameter values of the arcuate fasciculus of patients and control subjects (*n* = 15).

	Left AF				Right AF			
Patient	FA	TV	MD (×10^−3^ mm^2^/s)	FN	FA	TV	MD(×10^−3^ mm^2^/s)	FN
1	0.4912 **	986 **	0.7948	184 **	0.5285	3102	0.7013	479
2	0.4915 **	1975	0.7496	289 **	0.5526	2530	0.6962	580
3	0.4909 **	2435	0.7503	468	0.5233	2218	0.7369	444
4	0.4135 **	1912 **	0.8345 **	355	0.479	3240	0.7551	594
5	0.5162	831 **	0.7844	95 **	0.4754	2115	0.7705	384
Controls	0.5252 (0.015)	2551.625 (272.461)	0.7513 (0.026)	545.125 (89.606)	0.5200 (0.034)	2690.063 (429.588)	0.7233 (0.028)	524.750 (81.271)

Note: AF, arcuate fasiculus; FA, fractional anisotropy; TV, tract volume; FN, fiber number; MD, mean diffusivity. ** Parameters were significant difference compared with normal control subject values.

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
