# Peer review of "Structural Characteristic of the Arcuate Fasciculus in Patients with Fluent Aphasia Following Intracranial Hemorrhage: A Diffusion Tensor Tractography Study"

_brainsci, 2020, doi:10.3390/brainsci10050280_

Round 1
Reviewer 1 Report
The authors investigate the Diffusion Tensor Imaging profile of arcuate fasciculi in patients with fluent aphasia after intracerebral hemorrhage. It is a small case series but it has interesting findings. First off, the manuscript needs a thorough linguistic revision (I highlighted some of the sentences in the abstract). I have listed my other comments below.
Title
The title is wordy and not very clear. "According to Hemorrhagic Lesions" is redundant. Also, you looked at bilateral AF. I also recommend removing "the Left".
Abstract
Please reword "ICH lesions of patients 1 and 2 were placed on the left parietal lobe and the left basal ganglia" into ICH was located in..."
Reword " Those of patients 3, 4, and 5 were the left anterior temporal, the left temporal lobe, and the left frontal lobe, respectively".
Again reword it so that it is grammatically accurate. "Patients assessed language function using K-WAB (the Korean version of the Western Aphasia Battery) and reconstructed the AF using DTI. "
This following sentence needs rewording as well. " In DTI parameters of all patients, neural tract injury was shown (the decrement of FA, TV, and FN and increment of MD)".
I guess this following sentence is the conclusion in the abstract "Injury of the left AF is associated with aphasia, and the structural change of the left AF according to the hemorrhage in patients with various types of fluent aphasia following ICH is found using DTI." You need to elaborate on the sentence so that the conclusion you are reaching is clear to the reader.
Introduction
"In this study, we attempted to investigate the relationship between the structural characteristic of the left AF reconstructed using DTI and type of fluent aphasia according to hemorrhage lesion in subacute stroke patients with fluent aphasia following intracranial hemorrhage"
Again, you quantified bilateral AF. Also, you keep repeating "according to hemorrhage lesion". It is not clear to me what you meant by that.
Methods/Results
Please provide hematoma volumes as you are talking about the dislocated tracts. It would be beneficial if the dislocation happens with hematoma volumes higher than a certain limit of the hematoma volumes.
Any information about the etiology of the ICHs? It would be helpful to include them.
Discussion
Your results show the rightward lateralization of the microstructural integrity in AF. This was previously shown in another paper (PMID: 31884909) how after left hemispheric stroke, there is rightward lateralization in the AF. This needs a section in the discussion. Even, the degree of lateralization might be prognostically an important value.
Author Response
The authors investigate the Diffusion Tensor Imaging profile of arcuate fasciculi in patients with fluent aphasia after intracerebral hemorrhage. It is a small case series but it has interesting findings. First off, the manuscript needs a thorough linguistic revision (I highlighted some of the sentences in the abstract). I have listed my other comments below.
1. Title
The title is wordy and not very clear. "According to Hemorrhagic Lesions" is redundant. Also, you looked at bilateral AF. I also recommend removing "the Left".
Ans>
Your comment is proper
- As your mention, We remove “ left’ and “ according to hemorrhagic lesions”.
- we remove “ according to ~~” in the other
2. Abstract
Please reword "ICH lesions of patients 1 and 2 were placed on the left parietal lobe and the left basal ganglia" into ICH was located in..."
Ans>
- ICH lesions of patients 1 and 2 were placed on the left parietal lobe and the left basal ganglia" --> “ICHs of patients 1 and 2 were located in the left parietal ~~~”
-Reword " Those of patients 3, 4, and 5 were the left anterior temporal, the left temporal lobe, and the left frontal lobe, respectively".
Ans>
We change that sentence to “ICH was located in the left anterior temporal of patient 3, the left temporal lobe of patient 4, and the left frontal lobe of patient 5.”
-Again reword it so that it is grammatically accurate. "Patients assessed language function using K-WAB (the Korean version of the Western Aphasia Battery) and reconstructed the AF using DTI. "
Ans>
We change the sentence as below.
Patients assessed language function using K-WAB (the Korean version of the Western Aphasia Battery) and reconstructed the AF using DTI.-->
“We Patients assessed patients’ language function using K-WAB (the Korean version of the Western Aphasia Battery) and reconstructed the AF using DTI.”
- This following sentence needs rewording as well. " In DTI parameters of all patients, neural tract injury was shown (the decrement of FA, TV, and FN and increment of MD)".
Ans>
We change the sentence as below.
In DTI parameters of all patients, neural tract injury was shown (the decrement of FA, TV, and FN and increment of MD) --> “All patients showed neural tract injury (the decrement of FA, TV, and FN and increment of MD).”
- I guess this following sentence is the conclusion in the abstract "Injury of the left AF is associated with aphasia, and the structural change of the left AF according to the hemorrhage in patients with various types of fluent aphasia following ICH is found using DTI." You need to elaborate on the sentence so that the conclusion you are reaching is clear to the reader.
Ans>
Your comment is right
We change the sentence as below.
Injury of the left AF is associated with aphasia, and the structural change of the left AF according to the hemorrhage in patients with various types of fluent aphasia following ICH is found using DTI.--> “The structural difference of the AF in patients following ICH in the left hemisphere was associated with various types of fluent aphasia.”
3. Introduction
- "In this study, we attempted to investigate the relationship between the structural characteristic of the left AF reconstructed using DTI and type of fluent aphasia according to hemorrhage lesion in subacute stroke patients with fluent aphasia following intracranial hemorrhage"
Again, you quantified bilateral AF. Also, you keep repeating "according to hemorrhage lesion". It is not clear to me what you meant by that.
Ans>
We change the sentence as below.
“In this study, we attempted to investigate the relationship between the structural characteristic of the left AF reconstructed using DTI and type of fluent aphasia according to hemorrhage lesion in subacute stroke patients with fluent aphasia following intracranial hemorrhage"--> In this study, we attempted to investigate the relationship between the structural difference of bilateral AFs reconstructed using DTI and types of fluent aphasia in subacute stroke patients following intracranial hemorrhage (ICH) in the left hemisphere.
4. Methods/Results
- Please provide hematoma volumes as you are talking about the dislocated tracts. It would be beneficial if the dislocation happens with hematoma volumes higher than a certain limit of the hematoma volumes.]
Ans>
Your comment is right
We change the sentence as below.
We measured volume of hematoma
In Methods paragraph, we add the sentence as “We measured the volume of hematoma using a picture archiving communication system (PACS) on T2-weighted MRI images with reference to the previous study [31].”
In a result of each patient, the value of the volume of hematoma was added.
Patient 1, 27.44mL; Patient 2, 13.23mL; Patient 3, 11.87mL; Patient 4, 7.99mL; Patient 5, 11.73mL.
- Any information about the etiology of the ICHs? It would be helpful to include them.
Ans>
We change the sentence as below
We add the sentence for the etiology
Patient 1: A 60-year-old female with a history of hyperlipidemia
Patient 2: He did not have the past medical history.
Patient 3: A 50-year-old male without the past medical history
Patient 4: A 60-year-old female with a history of hypertension
Patient 5: A 58-year-old male without the past medical history
5. Discussion
- Your results show the rightward lateralization of the microstructural integrity in AF. This was previously shown in another paper (PMID: 31884909) how after left hemispheric stroke, there is rightward lateralization in the AF. This needs a section in the discussion. Even, the degree of lateralization might be prognostically an important value.
Ans> Your comment is right
We add the sentence as below
“It was suggested that the prognosis of aphasia in post-stroke patients was resulted from the structural change in the previous study [32]. This study showed that a right lateralization of AF led to poorer naming recovery. In constituent with the previous study, our patients (patients 1 and 2) found a lateralization and they showed Wernicke’s aphasia with poor naming function. However, while post-stroke patients in the previous study in the chronic stage, our patents were subacute stage. Thus, the further evaluation for long term follow up image and degree of lateralization is required.”
6. If you want to edit English by native speaker, we have the plan to edit English in MDPI.
Reviewer 2 Report
Unfortunately, I cannot recommend publication of this manuscript. The writing is so poor that it is basically impossible to assess the quality of the study in terms of its rationale, methods and results. Most statements are imprecise, unclear, or clearly wrong to give examples ‘Broca’s territory of the left AF was moved posteriorly based on that of the right AF’, ‘FA values indicated degrees of directionality at a microscopic level and microstructural integrity of axons, myelin, and microtubules’, ‘Patients assessed language function using K-WAB (the 18 Korean version of the Western Aphasia Battery) and reconstructed the AF using DTI’. Additionally, ‘fiber numbers’ was included as a parameters measurable by DTI, which makes me suspicious about whether the authors know what they are measuring with their methods. Tract delineation in itself is currently a very controversial topic, as it is sequence-, preprocessing and operator-dependent. Additionally, the arcuate fasciculus is a tract with large known inter-hemispheric differences. Delineating it in 5 patients, and comparing/interpreting the anatomy between left and right hemisphere does not seem like a sound scientific approach to me to understand its contribution in aphasia.
Author Response
Thank you for your great comments
I am sorry to confuse you. Your comment is right. However, I would like to explain and revise my manuscript in the detail about your comments.
1. The writing is so poor that it is basically impossible to assess the quality of the study in terms of its rationale, methods and results. Most statements are imprecise, unclear, or clearly wrong to give examples ‘Broca’s territory of the left AF was moved posteriorly based on that of the right AF’, ‘FA values indicated degrees of directionality at a microscopic level and microstructural integrity of axons, myelin, and microtubules’, ‘Patients assessed language function using K-WAB (the 18 Korean version of the Western Aphasia Battery) and reconstructed the AF using DTI’.
Ans>
- “FA values indicated degrees of directionality at a microscopic level and microstructural integrity of axons, myelin, and microtubules” means that because FA is usually useful to represent neural fiber injury and we indicated the mean of FA with the reference to the below studies
Reference:
- Jang, S.H.; Seo, J.P. Differences of the medial lemniscus and spinothalamic tract according to the cortical termination areas: A diffusion tensor tractography study. Somatosens. Mot. Res. 2015, 32, 67–71.
- Santillo, A.F.; Mårtensson, J.; Lindberg, O.; Nilsson, M.; Manzouri, A.; Landqvist Waldö, M.; van Westen, D.; Wahlund, L.O.; Lätt, J.; Nilsson, C. Diffusion Tensor Tractography versus Volumetric Imaging in the Diagnosis of Behavioral Variant Frontotemporal Dementia. PLoS One 2013, 8, 1–9.
-For neural tract injury, FA, MD, TV, and FN parameters were used. The reference presented that if one, two, or others of these parameters was abnormal compared with controls, there was the neural tract injury. Thus we defined the abnormality when one, two, or others of these parameters was abnormal compared with controls with the reference to the below reference.
Reference:
- Jang, S.H.; Lee, J.; Yeo, S.S. Central post-stroke pain due to injury of the spinothalamic tract in patients with cerebral infarction: A diffusion tensor tractography imaging study. Neural Regen. Res. 2017, 12, 2021–2024.
2. Patients assessed language function using K-WAB (the 18 Korean version of the Western Aphasia Battery) and reconstructed the AF using DTI’
Ans>
- the assessment for aphasia usually use WAB, and there is the Korean version (K-WAB). The previous study suggested the relationship between K-WAB and reconstruction of AF. The previous study is the blow
Thus, we revised the sentence in Line 72 “The assessment of language deficit used the K-WAB (the Korean version of the Western Aphasia Battery) as the previous study [3].”
Reference:
Kim, S.H.; Jang, S.H. Prediction of aphasia outcome using diffusion tensor tractography for arcuate fasciculus in stroke. Am. J. Neuroradiol. 2013, 34, 785–790.
Tak, H.J.; Jang, S.H. Relation between aphasia and arcuate fasciculus in chronic stroke patients. BMC Neurol. 2014, 14, 1–5.
3. fiber numbers’ was included as a parameters measurable by DTI, which makes me suspicious about whether the authors know what they are measuring with their methods. Tract delineation in itself is currently a very controversial topic, as it is sequence-, preprocessing and operator-dependent.
- Fiber number is one of parameters to represent neural tract injury. The previous study suggested that the FN value indicates the number of voxels included in a neural tract, thereby suggesting the total number of fibers within that tract, and significant decrement of the FN is neural tract injury. Thus, we add the below reference.
Reference:
Jang, S.H.; Seo, Y.S. Diagnosis of complex regional pain syndrome I following traumatic axonal injury of the corticospinal tract in a patient with mild traumatic brain injury. Diagnostics 2020, 10.
4. the arcuate fasciculus is a tract with large known inter-hemispheric differences. Delineating it in 5 patients, and comparing/interpreting the anatomy between left and right hemisphere does not seem like a sound scientific approach to me to understand its contribution in aphasia.
Ans>
Your comment is right. We also had a lot of concerns like your comment with the below reference. And the anatomical importance of the AF was added to the conclusion.
Thus, we add the sentence in limitation.
Reference:
Takaya, S.; Kuperberg, G.R.; Liu, H.; Greve, D.N.; Makris, N.; Stufflebeam, S.M. Asymmetric projections of the arcuate fasciculus to the temporal cortex underlie lateralized language function in the human brain. Front. Neuroanat. 2015, 9, 1–12.
Forkel, S.J.; De Schotten, M.T.; Dell’Acqua, F.; Kalra, L.; Murphy, D.G.M.; Williams, S.C.R.; Catani, M. Anatomical predictors of aphasia recovery: A tractography study of bilateral perisylvian language networks. Brain 2014, 137, 2027–2039.
If you want to edit English by native speaker, we have the plan to edit English in MDPI.